# Nanofiber Space-Confined Fabrication of High-Performance Perovskite Films for Flexible Conversion of Fluorescence Quantum Yields in LED Applications

**DOI:** 10.3390/polym16182563

**Published:** 2024-09-11

**Authors:** Ningbo Yi, Xue Guan, Xiaoting Chen, Luojia Xie, Nan Zhang, Jinfeng Liao, Long Su, Yancheng Wu, Feng Gan, Guoqiang Chang, Liyong Tian, Yangfan Zhang

**Affiliations:** 1College of Textile Science and Engineering, Wuyi University, Jiangmen 529020, China; yiningbo@wyu.edu.cn (N.Y.); gxykxy611@163.com (X.G.); upting-chen@outlook.com (X.C.); 15625631178@163.com (L.X.); longsu.81102@gmail.com (L.S.); yancheng_wu@126.com (Y.W.); gf@dhu.edu.cn (F.G.); 2Guangdong Provincial Key Laboratory of Semiconductor Optoelectronic Materials and Intelligent Photonic Systems, Harbin Institute of Technology, Shenzhen 518055, China; nanzhang@xjtu.edu.cn; 3Kai Rong De (Shao Guan) Fibre Glass Co., Ltd., Shaoguan 512000, China; changguoqiang@kingboard.com; 4MOE Key Laboratory for Nonequilibrium Synthesis and Modulation of Condensed 5 Matter, School of Physics, Xi′an Jiaotong University, Xi′an 710049, China; 5Joint Key Laboratory of the Ministry of Education, Institute of Applied Physics and Materials Engineering, University of Macau, Macau 999078, China; jinfengliao@um.edu.mo

**Keywords:** spatial confinement, perovskite, flexibility

## Abstract

Perovskite is an advanced optoelectronic semiconductor material that has garnered significant attention in recent years. However, its drawback lies in its environmental instability, limiting its practical applications. To tackle this issue, this research delved into the idea of creating a space-confined structure and used electrospinning to produce a film of perovskite nanocomposite fibers. By effectively encapsulating perovskite nanocrystals into a polymer matrix, the perovskite could be shielded from water and oxygen in the environment, thereby reducing the likelihood of perovskite decomposition and enhancing the stability of its structure and properties. This study examined the influence of material composition and the spinning process on the nanofiber structure to create good spatial confinement. This strategy resulted in a high photoluminescence quantum yield of over 80% and a long-term environmental stability of as long as 1000 h over 90% of the original PLQY. By harnessing the flexibility of the composite fibers, this study demonstrated the potential applications and performance of this nanocomposite film in flexible quantum fluorescence conversion for LED applications.

## 1. Introduction

In recent decades, the metal halide perovskite has been demonstrated to be a potential optoelectronic semiconductor material in solar cells [1,2], perovskite light-emitting diodes (LEDs) [3,4], photodetectors [5], and optical gain media for lasing [6]. It also could be a good quantum conversion medium to apply to display and color filtration technologies [7,8]. However, moisture and structural integrity are key factors that affect the long-termed commercial application of this material in industry [9]. Its nanostructure can be constructed in a facile crystal form with high quality and promote photoluminescence quantum yield (PLQY) efficiently due to the quantum confinement effect originating from the increase in exciton binding energy because of its size reduction [10,11]. To obtain high-quality quantum dots, the perovskite’s chemical constitution and synthesis process should be controlled during this study. The hot injection process was first developed to prepare perovskite quantum dots via the optimization of their temperature and additives (octadecene, oleic acid, etc.) under nitrogen conditions and could realize PLQYs of 50–90% corresponding to a nano size distribution of 12–42 nm [12]. During hot injection, the nanostructure’s size is not uniform, leading to a wide range of PLQYs due to temperature perturbations. The critical conditions of nitrogen and a high temperature also endow the delicate crystal structure with complexity in large-scale production. The ligand-assisted re-precipitation strategy has been demonstrated to obtain brightly luminescent and color-tunable colloidal CH_3_NH_3_PbX_3_ quantum dots with absolute quantum yields of up to 70% and average diameters of 3.3 nm, which can be adapted to perovskite nanoplates and nanorods [13]. This strategy can reduce the operation temperature and generate low-dimensional nanoscale quantum dots, but it is hard to obtain a uniform solid state from multiple batches of suspension liquid. A droplet-based microfluidic platform was adopted to prepare perovskite nanocrystals via the careful control of the reaction parameters, including the precursors′ stoichiometric proportions, reaction temperatures, and reaction times. The microfluidic system could be used to achieve CsPbX_3_ nanocrystals with average sizes ranging from 5 to 10 nm, demonstrating a new type of one-dimensional (1D) spatial confinement that could be used to prepare high-quality perovskite materials [14]. In previous research, embedding perovskite into several matrixes, such as hydrophobic polymers, multilayered structures, metal/organic frames, and core/shell structures, was demonstrated to be an efficient method for preventing its ionic degradation due to contact with environmental hydrogen [15]. To address the stability issue of perovskite, perovskite embedded into a polymer matrix could maintain its crystal structure effectively and generate high-performance devices. When composited with 3-aminopropyl triethoxysilane, perovskite nanocrystals are protected by a Si–O–Si protective layer exhibiting a long fluorescence lifetime and high quantum yield of more than 90% for more than 10 days [16]. A bulk heterostructure of quasi-2D/3D perovskites and a wide-optical-gap polymer exhibited external quantum efficiencies of up to 20.1% and a good stability of T50 = 46 h in air [17]. The polymer matrix not only experienced a performance enhancement, but it also provided protection for the perovskite against environmental destruction. These polymers enable perovskite materials to be processed in multiple ways to generate nanostructures like nanofibers and nanorods in films. One-dimensional nanostructures from micro-fluids could provide a strategy for producing uniform space-confined perovskite with high performance and stability characteristics. Therefore, developing a simple method for preparing one-dimensional quantum-confined perovskite is an important direction to promote the practical application and industrialization of high-performance perovskite materials. In addition, a perovskite structure with quantum confinement has a smaller volume size, making its optical and electronic properties more sensitive to structural changes, which is beneficial for enhancing the material′s optoelectronic performance. Electrospinning is a universal technique for preparing nanofiber membranes in many fields, like biomedical engineering, energy, environmental science, and electronic engineering. Nanofibers can be obtained on a large scale with good uniformity, which may overcome the obstacle of large-scale production using microfluidics. By combining the advantages of polymer matrixes and electrospinning, a quantum conversion film of a perovskite-based composite with a high PLQY and stability can be realized.

In this study, a quantum conversion membrane (QCM) was achieved with a high PLQY and good environmental stability via an electrospinning process based on the spatial confinement of polymers. This film was constructed with nanofibers composed of perovskite and polyacrylonitrile (PAN), which showed an emission wavelength of approximately 532 nm and a high conversion ability. In this study, the polymer matrix not only provided a matrix for the electrospinning process, but also maintained its structural integrity and provided protection for the perovskite so that it could obtain its long-term PL property. By optimizing the molar ratio of MABr and PbBr_2_ in the precursor spinning solution, the surface of the perovskite crystal could be passivized to enhance its stability and improve its fluorescence performance in the nanofibers. The content of perovskite in precursor spinning solution was investigated to obtain a high photoluminescence property due to the better confinement and resistance to environmental water molecules of a one-dimensional fibrous polymer matrix. The perovskite/PAN nanofiber film could be restored in RH 35% over 1200 h with an averaged PLQY retention of 90%. In the study of the key factor, voltage, the highest PLQY of the film was 80% at 22.5 kV, which was limited by the spinning machine. The nanofiber film resulting from this electrospinning method exhibited good flexibility, which could be used to demonstrate its potential as a quantum film in a flexible display. Therefore, electrospinning was demonstrated to be an effective strategy for obtaining a QCM with a high PLQY and stability for flexible display applications based on random 1D quantum-confined structures.

## 2. Materials and Methods

### 2.1. Materials

N, N-dimethylformamide (DMF, 99.5%, Aladdin, Shanghai, China), Dichloromethane (DCM, 99.97%, Macklin, Shanghai, China), Lead (II) Bromide (PbBr_2_, 99.99%, Macklin, Shanghai, China), Methyl ammonium bromide (MABr, MA^+^ = CH_3_NH_3_^+^, 99.5%, Xi′an p-OLED, Xi′an, China), and polyacrylonitrile (PAN, Mw ≈ 90,000, Aladdin, Shanghai, China) were all used as received without any further purification and were purchased from commercial sources.

Briefly, various molar ratios of MABr and PbBr_2_ (1.1:1, 1.5:1, 2:1, and 3:1) were added to 0.25 mL DMF and 0.1 mL DCM to create a perovskite precursor solution while stirring. Additionally, PAN powder was mixed with DMF to create a polymer solution with a concentration of 15 wt% while stirring at 50 °C for 3 h until it was fully dissolved. The perovskite (CH_3_NH_3_PbBr_3_) and PAN were added to the solution, with different solid contents of perovskite in the PAN matrix (6 wt%, 8 wt%, 10 wt%, and 12 wt%), to form the spinning precursor slurry. The spinning solution was then loaded into a 10 mL syringe. Different spinning voltages (15 kV, 17.5 kV, 20 kV, 22.5 kV, and 25 kV) were applied to produce various samples, allowing the perovskite crystal to be kept in different states during fiber formation and exhibit diverse performances. The injection rate was maintained at 0.8 mL/h to extrude the spinning solution from the syringe, with the fiber receiver positioned 20 cm away from the needle, resulting in a total spinning time of approximately 2 h. The membrane samples were utilized for subsequent tests and restored in a bag under a relative humidity of 35%.

### 2.2. Characterization

Absorption properties were analyzed using a UV-vis spectrophotometer (Shimadzu UV-2600, Tokyo, Japan). The photoluminescence quantum yield was measured with a fluorescence spectrometer (FLA4000, Hangzhou Jingfei Technology Co., Ltd., Hangzhou, China), using an integrating sphere method with an excitation wavelength of 365 nm. Photoluminescence spectra were recorded with a fluorescence spectrophotometer (RF-6000, Shimadzu, Kyoto, Japan). The samples’ morphology was tested with a fluorescence microscope (Olympus IX73, OLYMPUS, Tokyo, Japan). SEM and EDS mapping were performed to analyze the morphology and elements of the membranes (TESCAN Mira LMS, Brno, Czech Republic). XPS (Thermo Scientific K-Alpha+, Waltham, MA, USA) was utilized to investigate the surface distribution, depth profile, elemental composition, and valence state of their solid surfaces. A home-made flexible display was created using the QCM and featured an excitation backlight with a peak wavelength of 460 nm.

## 3. Results and Discussion

### 3.1. Preparation of QCMs

As mentioned above, the precursor spinning solution was prepared using a conventional method, by stirring at a temperature of approximately 50 °C, as illustrated in Figure 1a. It was crucial to add an amount of DCM into the perovskite solution to ensure rapid crystallization during spinning, with small crystals being embedded in the 1D fibrous polymer matrix with good spatial confinement. Previous studies indicated that an excess molar ratio of MABr could result in high-quality perovskite for both materials and devices used to repair the defects in polycrystals [18]. MABr and PbBr_2_ have varying solubilities in certain solvents, leading to defects in the crystal structure of perovskite during traditional spin coating. Thus, creating a different composition for the precursor solution used in printing and spinning could create high-quality perovskite crystals. Furthermore, it was shown that the polymer matrix could decrease the number of crystal defects due to the phase segregation of its complex hydrodynamics in a confined space, as reported. The polymer matrix utilized in this research needed to possess excellent environmental resistance to safeguard the perovskite. PAN was identified as a suitable polymer material with good weather resistance, sun resistance, stability at high temperatures, and the ability to retain 77% of its original strength after 18 months of outdoor exposure [19]. It also served as a versatile matrix for manufacturing fibers through various methods; in addition, it is resistant to chemical agents, inorganic acids, bleaching powder, hydrogen peroxide, and general organic agents [20]. PAN also exhibited a high hydrophobicity in its fiber state which provided a resistance to moisture for perovskite [21]. Therefore, PAN was selected as the fiber matrix and then combined with perovskite to ensure its long-term stability in diverse conditions. As depicted in Figure 1b, the composite membrane consisted of 1D perovskite/PAN fibers produced via electrospinning from a precursor mixture of PAN and Br–perovskite. In the TGA-DTA analysis of the perovskite/PAN composite film under an O_2_ atmosphere shown in Appendix A, the composition of the film exhibited good environmental stability below the temperature of 100 °C. The solid ratio of perovskite to PAN was carefully considered to generate smooth fibers, allowing the perovskite crystals to be fully embedded in the matrix. The membrane, as shown in Figure 1c, appeared white under natural light and efficiently emitted green light under excitation.

### 3.2. Structural and Optical Properties of QCMs

Crystal and structural integrity are key factors contributing to the properties of perovskite, affecting its emission efficiency depending on the drying method used. Electrospinning is a common method for preparing nanofibers, under high-voltage acceleration, from a non-Newtonian fluid. The fluidic PAN solidified and transformed into 1D fibers with the crystallization of perovskite in the matrix after the solvent evaporated during the electric field process. In Figure 1d, a photoluminescence peak around 521 nm and absorption below 510 nm could be observed, corresponding to the characteristic existence of nanosized Br–perovskite in the fiber film [22]. The structural analysis in XRD (Figure 1e) confirmed that the perovskite nanocrystals were well-distributed in a polycrystalline manner in the PAN matrix, as indicated by its typical indices (100), (110), (200), (210), (112), (220), and (300) [23]. The broad peak between 15° and 20° could be attributed to the orthorhombic reflection of the (110) and (200) indices of PAN, as previously reported [24,25]. The SEM images and element analysis in Figure 1b and Appendix A revealed that the crystal morphology was barely visible in the fiber at a high resolution, suggesting that the perovskite likely grew to a nano size due to the spatial constraints imposed by the 1D fiber structure during its formation. Therefore, the 1D fibrous structure formed during electrospinning offered the potential for growing perovskite nanocrystals and achieving a high photoluminescence quantum yield with good environmental stability under the protection of a polymer matrix.

### 3.3. Optimization and Characteristics of QCMs

#### Optimization of QCM Composition

Considering the stoichiometry of perovskite, a specific ratio of PbBr_2_ to MABr could be dissolved in the precursor solution and crystallized to form perovskite after drying. Previous reports have shown that different ratios can result in varying degrees of structural integrity due to the distinct crystallization rates of PbBr_2_ and MABr [18]. By adjusting the ratio of PbBr_2_ to MABr, high-quality perovskite films can be achieved in optoelectronic devices. Therefore, the initial precipitation of PbBr_2_ and MABr and their subsequent drying under an electric field were able to lead to the growth of a complete perovskite crystal during electrospinning. The photoluminescence quantum yield (PLQY) was a crucial parameter for evaluating the quality of the perovskite in the QCMs. As depicted in Figure 2a, the PLQY varied with the precursor component of perovskite in the polymer matrix. Under specific fabrication conditions, a ratio of 1:1.5 was considered an optimal choice for manufacturing QCMs to attain the highest PLQY. An excess amount of MABr could eliminate defects and passivate the perovskite phase to enhance performance, which was consistent with our previous study [26]. MABr not only repaired crystal defects during the spin-coating process of film preparation but also facilitated the formation of perovskite nanocrystals in 1D fibers during their relatively short drying time while electrospinning. In the SEM images shown in Figure 2b, smooth morphologies of perovskite-based fibers could be observed in various precursor solutions, indicating an absence of micrometer-sized crystals forming during their rapid drying while electrospinning. The PL images of the perovskite-based fibers in the QCMs validated that optimizing these components could result in continuous emission along the fiber, as shown in the inset in Figure 2b. To understand the chemical states of the main elements in the QCMs, we examined the elemental compositions of the QCMs from various perspectives. The survey scans in Figure 2c and Appendix A revealed a Br to Pb ratio of over 3, ensuring the integrity of the crystal structure of the perovskite in the fibers. However, this ratio varied in the precursor solution due to the shallow testing depth of about tens of nanometers from the surface of fibers. The XPS analysis of Br in Figure 2d identified peaks around 67.1 eV and 68.5 eV, corresponding to Br 3d_5/2_ and Br 3d_3/2_. The peaks in the sample with a ratio of 1.5:1 appeared narrow due to the purification of the chemical state in the composition of the fibers. Furthermore, the peaks in the XPS spectra of Pb also exhibited coherence with the Br in the same sample, with the peaks at around 137.4 and 142.3 eV being assigned to Pb 4f_7/2_ and 4f_5/2_, which was indicative of the presence of Pb^2+^ in the perovskite, as previously reported [27]. As the ratio of PbBr_2_ to MABr increased in Figure 2e, the multiple peaks of Pb 4f_7/2_ and 4f_5/2_, along with the broader peaks of Br 3d_5/2_ and Br 3d_3/2_, confirmed the presence of a structural perturbation in the 1D fiber system. By combining the XPS analysis of the chemical states of Br and Pb in different samples, it was observed that samples with purified and narrower peaks exhibited the highest PLQY values, such as the 1.5:1 ratio. This precursor ratio was selected following optimization and preparation.

The polymer matrix played a crucial role in the QCMs, enabling the processibility of the 1D nanofiber structure and serving as a protective coating for the perovskite nanocrystals. The perovskite content needed adjustment, impacting not only the viscosity for forming a Taylor cone during electrospinning but also determining the PL’s performance and stability based on the perovskite′s crystal size and distribution within the fibers. The photoluminescence property and its color coordinate were investigated, as shown in Appendix A, which showed that it is green with a large color gamut over (0.3000, 0.6000), as defined in the sRGB. The investigation of the perovskite contents of various QCMs was conducted using PLQY, and its retention was maintained under RH 35% conditions and a room temperature of approximately 25 °C. In Figure 3a, it was observed that the PLQY initially increased with the perovskite content, peaked at a ratio of 10%, and then decreased. The 10% content yielded the highest PLQY in the study. As the content increased, the perovskite helped reduce the excitation light wastage in a specific fiber system. However, excessive perovskite content led to a reduced emission ability due to the concentrated perovskite distribution in the fibers. When the content of perovskite reached a certain ratio, the perovskite then crystallized with increasing precursor content and formed large crystals during electrospinning. When the crystal size increased, the PL property of perovskite could result in a relatively low PLQY. The study of PLQY retention was crucial for assessing the stability of QCMs, a key reliability factor in their practical application. Moisture was identified as the primary factor contributing to the decline in PL performance due to the degradation of the crystal structure. The perovskite/PAN composite fiber in this research exhibited excellent hydrophobicity with a contact angle exceeding 150° in Figure 3b, providing resistance to water molecules and ensuring the high stability of the perovskite crystal structures. The investigation highlighted that a perovskite content of around 10 wt% could maintain a high PLQY retention level, as shown in Figure 3c, while an excessive amount could compromise stability due to inadequate protection from the PAN matrix in the confined 1D space. Furthermore, the QCMs developed in this study demonstrated a long-term PLQY retention exceeding 1000 h under ambient conditions, showcasing their potential as light conversion materials in practical applications.

The acceleration voltage applied during electrospinning was a crucial technological parameter for obtaining high-quality nanofibers. The diameter of the nanofiber could be determined by the voltage applied during electrospinning, which provided a large enough confined space to embed perovskite nanocrystals. In addition, the precursor components of the hybrid perovskite, as an ionic compound, could be sensitive to the voltage between the two electrodes, requiring a balance between accelerating the voltage and crystallization. Therefore, the voltage was studied in relation to the PLQY for optimization within the controllable range of the machine. As depicted in the SEM images in Appendix A, uniform and smooth fibers could be produced within the voltage range of 15–25 kV. By analyzing the PLQY in relation to voltages in Figure 3d, it was shown that a voltage of 22.5 kV could result in high-performance QCMs. This voltage struck a balance between the crystallization rate of perovskite in the polymer matrix and the production time of the fibers. By optimizing both the voltage and the perovskite content, the PLQY of the QCMs could potentially reach as high as 80%, indicating the good performance of perovskite in compositing with other polymers, as shown in Appendix A [28,29,30,31,32,33,34,35,36].

Light transformation is utilized in various fields to convert incident light into the desired color or wavelength based on the energy level structures of photoelectric conversion materials. QCM, a type of photoelectric conversion material known for its high efficiency and narrow peak width at half height, has become essential in areas such as in displays, scientific research, electronic information processing, and communication technology. Particularly significant in the realm of high-quality flexible displays, QCMs have played a crucial role in enhancing color brightness and expanding the range of displayable colors. Research here has highlighted the use of QCMs composed of perovskite/PAN nanofibers, providing a flexible solution for color conversion in display applications, as demonstrated in Figure 4a. In Figure 4b, the compact design of these QCMs, with a thickness of approximately 20 μm, emphasizes their exceptional conversion efficiency, ensuring the transformation of backlight into green light under excitations of different intensities. These innovative QCMs have achieved the remarkable ability to efficiently convert blue LED light on flexible substrates into green hues, whether in flat or curved configurations (Figure 4c) because of the good photoluminescence properties of perovskite-based QCMs. Additionally, this advancement in QCM technology holds great promise for revolutionizing the display industry and expanding the possibilities of color representation on flexible screens.

## 4. Conclusions

In this paper, a flexible QCM was manufactured with a high PLQY and stability using the facile method of electrospinning. SEM images showed that smooth sub-micrometer fibers could be prepared to create a 1D confined space for the growth of perovskite nanocrystals during spinning. This 1D confined space promoted a high PLQY of perovskite in the PAN matrix. By adjusting the ratio of MABr to PbBr_2_, the nanostructures of the perovskite could be optimized to eliminate defects and achieve good crystallization, with a good performance at a ratio of 1.5:1. Modulating the content of perovskite in the PAN matrix ensured the complete coating of the nanostructured perovskite by the polymer, preventing its environmental degradation due to water molecules and enabling long-term stability and applicability for perovskite at a content of 10 wt%. When the applied high voltage reached approximately 22.5 kV, QCMs with good performance properties could be obtained by balancing the formation of 1D fibers and the confinement of perovskite nanocrystals. This study demonstrated that a high PLQY of up to 80% could be achieved after optimizing the process and composition. The QCMs, which were composed of nanofibers, were shown to be promising films for use in displays as they efficiently convert blue light into green light while maintaining good flexibility. In conclusion, the QCMs produced through electrospinning represent a promising material and strategy for achieving light conversion in flexible light sources and displays.

## Figures and Tables

**Figure 1 polymers-16-02563-f001:**
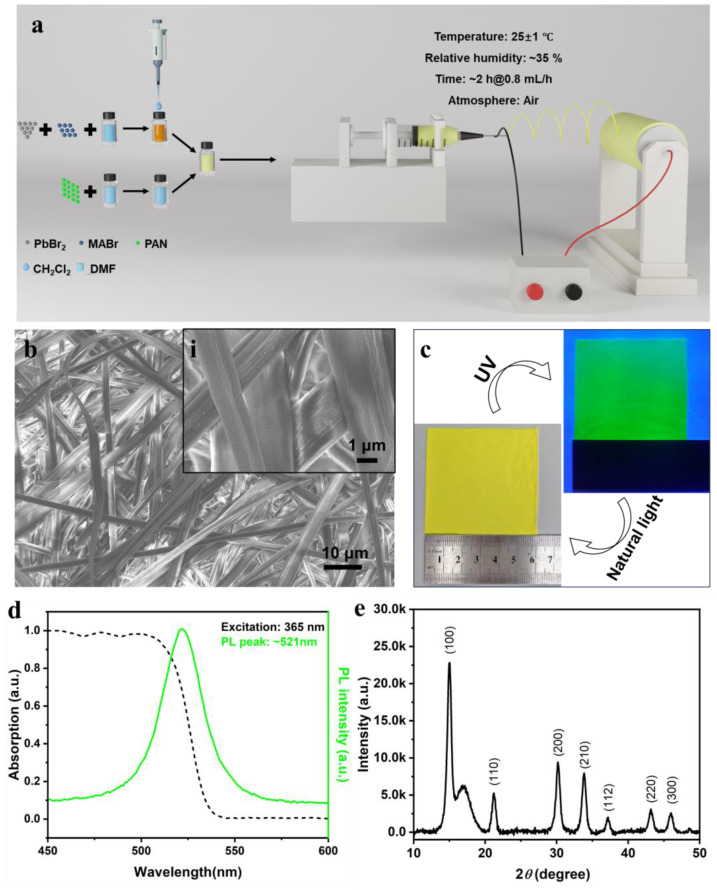
Preparation, structures, and optical properties of QCMs. (**a**) A schematic illustration of the preparation of QCMs from the precursor solution via electrospinning. (**b**) Fibrous structures in the QCMs. Inserted (i): SEM image of fiber structure with high resolution. (**c**) A photo of QCMs under natural light and UV light. (**d**) The absorption and PL properties of QCMs. (**e**) XRD structural analysis of the perovskite crystals in the fibers.

**Figure 2 polymers-16-02563-f002:**
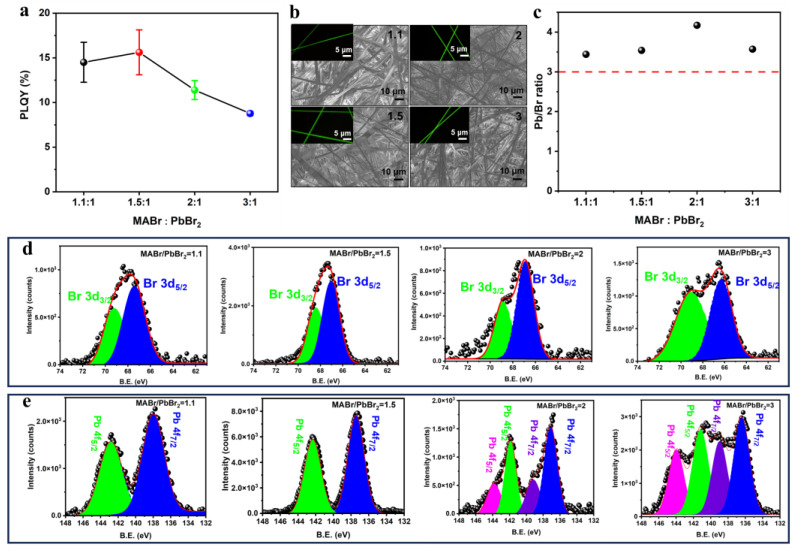
The chemical compositions of fibers with perovskite/PAN compositions. (**a**) The PLQY’s dependence on the MABr/PbBr_2_ ratio in the precursor solution. The PLQY of each ratio was tested from 5 samples and the average value was obtained as circle mark to avoid occasionality. (**b**) Fibrous structures from different precursor solutions with various ratios of MABr/PbBr_2_. The inserted PL microscope photograph corresponded to each sample. (**c**) The ratio of Br/Pb in different samples. (**d**) Br 3d spectra in different samples. (**e**) Pb 4f spectra in different samples.

**Figure 3 polymers-16-02563-f003:**
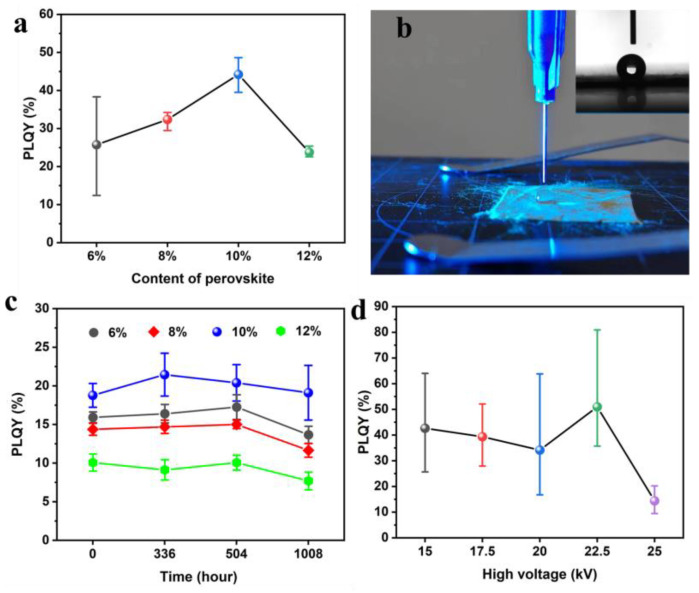
Effects of the perovskite content and process on its properties. (**a**) The PLQY’s dependence on the perovskite content.Circle in each color represented the average value in statistics from 5 samples. (**b**) The hydrophobic properties of the QCMs. Inserted image: contact angle testing. (**c**) The PLQY retention was restored in ambient environment with RH 35% and a room temperature of approximately 25 °C. (**d**) Analyzing the PLQY in relation to the high voltages applied in electrospinning, for which the highest PLQY reached 80%.Circle in each color represented the average value in statistics from 10 samples.

**Figure 4 polymers-16-02563-f004:**
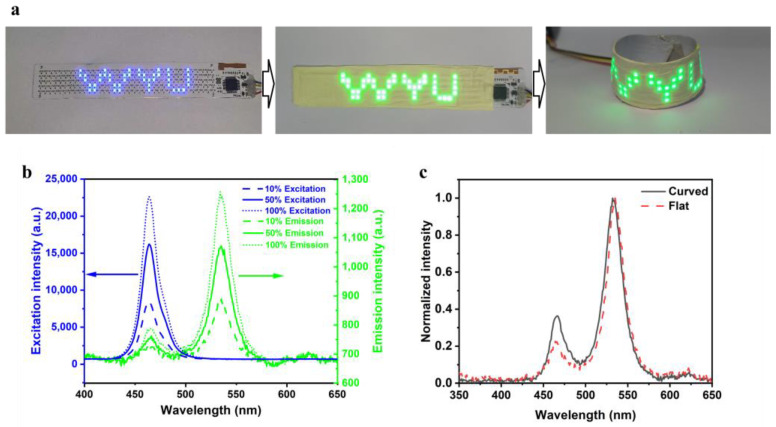
The properties and applications of QCMs. (**a**) A demonstration of color conversion in display applications. (**b**) The PL transformation of backlight into green light under the excitation of different intensity by QCMs. (**c**) Innovative QCMs achieving a remarkable PL ability with good flexibility.

## Data Availability

Data are contained within the article and Appendix A.

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
