# Peer review of "Nanofiber Space-Confined Fabrication of High-Performance Perovskite Films for Flexible Conversion of Fluorescence Quantum Yields in LED Applications"

_polymers, 2024, doi:10.3390/polym16182563_

Round 1

Reviewer 1 Report

Comments and Suggestions for Authors

The work presented in this article is interesting and the use of Perovskite crystals for LED applications in polymer matrix is reported. The materials are characterized by number of cahracterization techniques. The work is novel and can be accepted for publication after minor revision. 

1. What is the chemical formula of the perovskite crystal. It is not clear from the manuscript. 

2. Polymer with P-crystal characterizations must be improved. TG-DTA analysis must be included. 

3. Can authors provide CIE color coordinates for the PL studied? 

4. The Perovskite crystals / particles are not visible in the PAN polymer in Figure 1. Can authors provide SEM to prove the distribution of crystals in matrix. 

Author Response

Dear Reviewer, 

Thank you for your guidance with regard to our manuscript - we appreciate greatly for your work. We are also grateful for the valuable suggestions and comments. Based on the reviewers’ reports, we have carefully revised the manuscript and all of the comments have been addressed accordingly in the attachment.  

We hope that with these improvements you will now find the paper suitable for publication in Polymers.

Sincerely,

Ningbo Yi

Reviewer 2 Report

Comments and Suggestions for Authors

The manuscript by Ningbo Yi et cl. described a perovskite composite fiber using electrospinning method that involves encapsulating perovskite nanocrystals within a polymer matrix. The polymer shields the perovskite from moisture and oxygen, thereby reducing decomposition and improving stability. Fabrication conditions including molar ratio, perovskite content and electrospinning voltage were optimized to maximize the PLQY of the composite. The authors further demonstrated a flexible display based on this material that can transform blue backlight into green light under different excitation intensity. Overall, the manuscript is well-written, and the experiments were well designed and performed. However, the analysis can be enhanced to improve the scientific soundness. I recommend the manuscript to be published under the condition that the comments below are addressed.  

1.      I suggest the authors to change the fill color for PAN, DMF and CH2Cl2 in figure 1a. Those colors are too close to the background and are hard to distinguish.

2.      Why do authors term the fiber “nanofiber”, considering the width of the fiber is couple microns as shown in figure 1b (and insert).

3.      Why do authors claim that perovskite is in the form of nanocrystals? Just by stating perovskite crystal was barely visible under high resolution SEM (line 180 – 181) is not direct evidence. The authors might want to compare the emission or absorption energy to the reported values of bulk and nanosized MAPBBr3.

4.      Some statements lack references. E.g. line 196, “Previous reports have shown…” needs reference.

5.      Line 175, “photoluminescence and absorption spectra showed an emission peak and absorption located at 530 nm, corresponding to the characteristic bandgap of Br-perovskite.” Are the authors suggesting there is no stokes shift observed? This is rare for perovskite, typically there is an energy difference between absorption and emission peak. Just an example, in Scientific Reports 10, 15720 (2020) the absorption and emission of MAPbBr3 nanoparticles show tens of meV energy difference.

6.      Line 247 “As the content increased, the perovskite helped reduce excitation light wastage in a specific fiber system. However, excessive content led to reduced emission ability due to the concentrated perovskite distribution in the fibers.” Could the authors elaborate on these? Why is excitation light wastage affecting PLQY, as PLQY is only considering the emission vs the absorbed excitation. The total excitation power or absorption % is not a factor. Also, why is concentrated perovskite distribution lowering PLQY – is there a change in structure or defect density? The simple attribution here just lacks evidence.

7.      Why can the flexible display transform blue backlight into green light under different excitation intensity? This seems to suggest that the absorption or PLQY of the composite has power-dependence. Have the authors looked into this?

8.      Line 294 “..., thanks to the photoluminescence mechanism depicted in Figure 4c”. However, Figure 4c is the emission spectra of the QCM flexible screen, rather than mechanism illustration.

Author Response

(The authors gave the same response as above.)

Reviewer 3 Report

Comments and Suggestions for Authors

Authors should address these comments. 

Author Response

(The authors gave the same response as above.)
